# Thermal Ablation Compared to Partial Hepatectomy for Recurrent Colorectal Liver Metastases: An Amsterdam Colorectal Liver Met Registry (AmCORE) Based Study

**DOI:** 10.3390/cancers13112769

**Published:** 2021-06-02

**Authors:** Madelon Dijkstra, Sanne Nieuwenhuizen, Robbert S. Puijk, Florentine E.F. Timmer, Bart Geboers, Evelien A.C. Schouten, Jip Opperman, Hester J. Scheffer, Jan J.J. de Vries, Rutger-Jan Swijnenburg, Kathelijn S. Versteeg, Birgit I. Lissenberg-Witte, M. Petrousjka van den Tol, Martijn R. Meijerink

**Affiliations:** 1Department of Radiology and Nuclear Medicine, Amsterdam University Medical Centers, VU Medical Center Amsterdam, Cancer Center Amsterdam, 1081 HV Amsterdam, The Netherlands; s.nieuwenhuizen1@amsterdamumc.nl (S.N.); r.puijk@amsterdamumc.nl (R.S.P.); f.timmer1@amsterdamumc.nl (F.E.F.T.); b.geboers@amsterdamumc.nl (B.G.); e.schouten@amsterdamumc.nl (E.A.C.S.); hj.scheffer@amsterdamumc.nl (H.J.S.); j.devries1@amsterdamumc.nl (J.J.J.d.V.); mr.meijerink@amsterdamumc.nl (M.R.M.); 2Department of Radiology and Nuclear Medicine, Noordwest Ziekenhuisgroep, location Alkmaar, 1800 AM Alkmaar, The Netherlands; j.opperman@nwz.nl; 3Department of Surgery, Amsterdam University Medical Centers, VU Medical Center Amsterdam, Cancer Center Amsterdam, 1081 HV Amsterdam, The Netherlands; r.j.swijnenburg@amsterdamumc.nl (R.-J.S.); mp.vandentol@amsterdamumc.nl (M.P.v.d.T.); 4Department of Medical Oncology, Amsterdam University Medical Centers, VU Medical Center Amsterdam, Cancer Center Amsterdam, 1081 HV Amsterdam, The Netherlands; k.versteeg@amsterdamumc.nl; 5Department of Epidemiology and Data Science, Amsterdam University Medical Centers VU Medical Center Amsterdam, Vrije Universiteit Amsterdam, 1081 HV Amsterdam, The Netherlands; b.lissenberg@amsterdamumc.nl

**Keywords:** colorectal liver metastases (CRLM), recurrence, thermal ablation, partial hepatectomy (PH), microwave ablation (MWA), radiofrequency ablation (RFA), repeat local treatment

## Abstract

**Simple Summary:**

Between 64 and 85% of patients with colorectal liver metastases (CRLM) develop distant intrahepatic recurrence after curative intent local treatment. The current standard of care for new CRLM is repeat local treatment, comprising partial hepatectomy and thermal ablation. Although relatively safe and feasible, repeat partial hepatectomy can be challenging due to adhesions and due to the reduced liver volume after surgery. This AmCORE based study assessed safety, efficacy and survival outcomes of repeat thermal ablation as compared to repeat partial hepatectomy in patients with recurrent CRLM. Repeat partial hepatectomy was not different from repeat thermal ablation with regard to survival, distant- and local recurrence rates and complications, whereas length of hospital stay favored repeat thermal ablation. Thermal ablation should be considered a valid and potentially less invasive alternative in the treatment of recurrent new CRLM, while the eagerly awaited results of the COLLISION trial (NCT03088150) should provide definitive answers regarding surgery versus thermal ablation for CRLM.

**Abstract:**

The aim of this study was to assess safety, efficacy and survival outcomes of repeat thermal ablation as compared to repeat partial hepatectomy in patients with recurrent colorectal liver metastases (CRLM). This Amsterdam Colorectal Liver Met Registry (AmCORE) based study of two cohorts, repeat thermal ablation versus repeat partial hepatectomy, analyzed 136 patients (100 thermal ablation, 36 partial hepatectomy) and 224 tumors (170 thermal ablation, 54 partial hepatectomy) with recurrent CRLM from May 2002 to December 2020. The primary and secondary endpoints were overall survival (OS), distant progression-free survival (DPFS) and local tumor progression-free survival (LTPFS), estimated using the Kaplan–Meier method, and complications, analyzed using the chi-square test. Multivariable analyses based on Cox proportional hazards model were used to account for potential confounders. In addition, subgroup analyses according to patient, initial and repeat local treatment characteristics were performed. In the crude overall comparison, OS of patients treated with repeat partial hepatectomy was not statistically different from repeat thermal ablation (*p* = 0.927). Further quantification of OS, after accounting for potential confounders, demonstrated concordant results for repeat local treatment (hazard ratio (HR), 0.986; 95% confidence interval (CI), 0.517–1.881; *p* = 0.966). The 1-, 3- and 5-year OS were 98.9%, 62.6% and 42.3% respectively for the thermal ablation group and 93.8%, 74.5% and 49.3% for the repeat resection group. No differences in DPFS (*p* = 0.942), LTPFS (*p* = 0.397) and complication rate (*p* = 0.063) were found. Mean length of hospital stay was 2.1 days in the repeat thermal ablation group and 4.8 days in the repeat partial hepatectomy group (*p* = 0.009). Subgroup analyses identified no heterogeneous treatment effects according to patient, initial and repeat local treatment characteristics. Repeat partial hepatectomy was not statistically different from repeat thermal ablation with regard to OS, DPFS, LTPFS and complications, whereas length of hospital stay favored repeat thermal ablation. Thermal ablation should be considered a valid and potentially less invasive alternative for small-size (0–3 cm) CRLM in the treatment of recurrent new CRLM. While, the eagerly awaited results of the phase III prospective randomized controlled COLLISION trial (NCT03088150) should provide definitive answers regarding surgery versus thermal ablation for CRLM.

## 1. Introduction

Colorectal cancer (CRC) is the third most common form of cancer worldwide [1]. Up to 50% of patients develop colorectal liver metastases (CRLM), a lethal condition in the vast majority of cases [2,3]. The only chance for cure entails a radical intent treatment of the CRLM, including partial hepatectomy and/or thermal ablation (i.e., radiofrequency ablation (RFA), microwave ablation (MWA)) [4]. Although the 5-year overall survival (OS) nowadays reaches 50–60% [5,6], only 20% of patients with CRLM are eligible for curative intent treatment.

In the past few decades surgical resection has been considered the gold standard in upfront resectable CRLM, while thermal ablation emerged for small (≤ 3 cm) unresectable CRLM [3,7,8,9,10]. When compared to partial hepatectomy, thermal ablation is currently associated with a lower complication rate, reduced hospital stay and lower costs but also with an inferior survival according to two recent meta-analyses and propensity score analyses [3,10,11,12,13,14]. Given the high risk of selection bias when comparing partial hepatectomy for resectable tumors with thermal ablation for unresectable disease, survival outcomes of the two techniques are currently considered to be in equipoise and the results of the prospective COLLISION trial (NCT03088150) are eagerly awaited [8]. Although curative intent local treatment offers complete tumor eradication in most, 64–85% of patients develop new metastases, commonly detected within 12 months following the initial treatment [15,16,17], of which the liver is the sole site of recurrence in approximately 39–43% [18].

Large international multi-institutional retrospective series and several other groups on repeat partial hepatectomy with curative intent of new CRLM demonstrated 5-year OS following the second treatment reaching 51% [19,20,21,22]. As a result the current standard of care for new CRLM is repeat local treatment, either upfront or after induction chemotherapy [23,24,25,26,27,28]. Although relatively safe and feasible, repeat partial hepatectomy can be challenging due to adhesions and due to the reduced liver volume after surgery [29]. Given its superior safety profile and the fact that thermal ablation is less affected by previous surgical injury, the question has arisen whether thermal ablation could be a safer and equally effective alternative to repeat partial hepatectomy for small-size recurrences [30].

This Amsterdam Colorectal Liver Met Registry (AmCORE) based study aimed to analyze safety, efficacy and survival outcomes following repeat thermal ablation compared to repeat partial hepatectomy for recurrent CRLM. 

## 2. Materials and Methods

This single-center prospective cohort study was performed at the Amsterdam University Medical Centers—location VU Medical Center, the Netherlands, a tertiary referral center for hepatobiliary and gastrointestinal malignancies. Data were extracted from the AmCORE prospectively maintained CRLM database. The study was approved by the affiliated Institutional Review Board (METc VUmc: 2021.0121). The analyzed study data reported conform to the ‘Strengthening the Reporting of Observational studies in Epidemiology’ (STROBE) guideline [31].

### 2.1. Patient Selection

Data of all patients with recurrent new CRLM after initial curative intent local treatment, eligible for repeat local treatment were collected from the prospective database. Additional recollecting of data was performed by retrospectively searching the hospital’s electronic patient database when required. Patients undergoing repeat thermal ablation or repeat partial hepatectomy were included. Patients with loss to follow-up or undergoing repeat stereotactic body radiation therapy (SBRT), irreversible electroporation (IRE) or a combination of resection and thermal ablation in the same procedure, were excluded. 

### 2.2. Repeat Local Treatment Procedures

Recurrent new CRLMs were detected during follow-up using cross-sectional imaging containing contrast enhanced computed tomography (ceCT) and ^18^F-fluoro-2-deoxy-D-glucose (^18^F-FDG) positron emission tomography (PET)-CT scans, using contrast enhanced magnetic resonance imaging (ceMRI) with diffusion-weighted images prior to repeating local treatment. The choice of the repeat local treatment procedure was based on local expertise, determined by multidisciplinary tumor board evaluations attended by (interventional) radiologists, oncological or hepatobiliary surgeons, medical oncologists, radiation oncologists, nuclear medicine physicians, gastroenterologists and pathologists. Repeat local treatment was performed by an experienced interventional radiologist (mastery degree in image-guided tumor ablation, having performed and/or supervised >100 thermal ablation procedures) or by an experienced, certified oncological or hepatobiliary surgeon (with broad expertise, having performed and/or supervised >100 liver tumor resection procedures). Resections were performed at discretion of the performing oncological or hepatobiliary surgeon, comprising the extent and specific technique as well as resection margins (with the intention and preoperative estimation of a possible pathological R0 resection). Metastectomy was performed when eligible to preserve liver volume and anatomical resection when necessary. Thermal ablation procedures were performed at the discretion of the interventional radiologist, according to the CIRSE quality improvement guidelines (with an intentional tumor free ablation margin >1 cm, confirmed with computational techniques and image fusion or estimated in the early years) [32]. Percutaneous approach was preferred in patients with no contra-indications (proximity of critical structures). When insufficiently ablated margins were presumed and/or confirmed by ceCT or ceMRI following thermal ablation, residual unablated tumor tissue was retreated with overlapping ablations. Conformal to national guidelines, (neo)adjuvant chemotherapy was not routinely administered, with the exception of cases where downsizing would likely reduce procedural risk (induction chemotherapy) or for patients with biologically unfavorable early multiple intrahepatic recurrences (<6 months following the initial treatment) [27]. 

### 2.3. Follow-Up

^18^F-FDG-PET-CT with diagnostic ceCTs of the chest and abdomen were performed in the first year 3/4-monthly, in the 2nd and 3rd year 6-monthly and in the 4th and 5th year 12 monthly after repeat local treatment, according to national guidelines [27]. CeMRI with diffusion-weighted images was used as problem solver. In the context of a presumably incomplete percutaneous ablation procedure, a ceCT-scan was performed within one to six weeks after the repeat local treatment. Local tumor progression (LTP) was defined as a solid and unequivocally enlarging mass or focal ^18^F-FDG PET avidity at the surface of the ablated tumor or resection margin, and histopathological confirmation in case of uncertainty.

### 2.4. Data Collection and Statistical Analysis

Patient and treatment characteristics were obtained from the AmCORE database. Categorical variables are reported as number of patients with percentages and continuous variables are reported as mean with standard deviation (SD) when normally distributed and as median with interquartile range (IQR) when not-normally distributed. The patients were divided into two groups regardless of initial treatment: repeat thermal ablation and repeat partial hepatectomy. Characteristics between groups were compared using the Fisher’s exact test for dichotomous variables, using the Pearson Chi square test for categorical variables and using independent samples t-test when normally distributed and Mann–Whitney U Test when not-normally distributed for continuous variables. 

Primary endpoint OS and secondary endpoints local tumor progression-free survival (LTPFS) and distant progression-free survival (DPFS) were defined as time-to-event from repeat local treatment. Death without local or distant progression (competing risk) was censored. Complications were described using Common Terminology Criteria for Adverse Events 5.0 (CTCAE) [33].

Primary endpoint OS was reviewed using the Kaplan–Meier method using the log-rank test and comparison between the two groups was conducted using Cox proportional hazards regression models, accounting for potential confounders in multivariable analysis. Secondary endpoint complications, LTPFS and DPFS were analyzed using the chi-square test and the Kaplan–Meier method using the log-rank test and Cox proportional hazards regression models to account for potential confounders. Variables with *p* < 0.100 in univariable analysis were included in multivariable analysis using forward selection procedure. Significant variables, *p* = 0.050, were reported as potential confounders and further investigated. Variables were considered confounders when the association between the two treatment groups and OS, DPFS, LTPFS differed >10% in the corrected model. Corrected hazard ratio (HR) and 95 per cent confidence interval (95% CI) were calculated. Length of hospital stay was analyzed using the Mann–Whitney U test. Subgroup analyses were performed to assess heterogeneous treatment effects according to patient, initial and repeat local treatment characteristics.

Statistical analyses, supported by a biostatistician (BLW), were performed using SPSS^®^ Version 24.0 (IBM^®^, Armonk, New York, NY, USA) [34] and R version 4.0.3. (R Foundation, Vienna, Austria) [35].

## 3. Results

After identification of patients with recurrent CRLM in the AmCORE database, 136 patients were selected for the analysis of recurrent CRLM, of which 100 were treated with repeat thermal ablation and 36 with repeat partial hepatectomy (Figure 1). A total of 224 tumors were treated with repeat ablation (*n* = 170) or repeat partial hepatectomy (*n* = 54) between May 2002 and December 2020. 

### 3.1. Patient Characteristics

Table 1 presents patient characteristics of the 136 included patients. There were no significant differences between the two treatment groups. The age ranged between 27 and 86 years. Median time between initial treatment and diagnosis of recurrence was 6.9 (IQR 4.0–13.4) months, 6.4 (IQR 4.0–10.4) months in the repeat thermal ablation group and 12.2 (IQR 3.7–21.3) in the repeat partial hepatectomy group (*p* = 0.056). Most patients had 1 recurrent CRLM (62.5%) and size of largest metastasis was mostly small (1–30 mm; 84.7%). Median follow-up time after repeat thermal ablation was 23.3 months and after repeat partial hepatectomy 34.9 months. Median tumor size was 21 (IQR 12.5–26.5) in the partial hepatectomy group and 16.5 (10.75–23.0) in the thermal ablation group.

### 3.2. Treatment Characteristics

Table 2 shows treatment characteristics of the procedures concerning type of system used for thermal ablation and partial hepatectomy (operation) technique. Comparison of local treatment method showed that the majority of the repeat thermal ablation group underwent a percutaneous approach and the majority of repeat partial hepatectomy group underwent an open approach. A total of 40 patients received treatment with RFA (40.0%), all prior to 2017, and 60 patients (60.0%) received treatment with MWA. In the partial hepatectomy group, the majority of patients received minor repeat resection (97.1%). Median length of hospital stay of the entire cohort was 1.0 days (IQR 1.0–3.3), of the repeat thermal ablation group 1.0 days (IQR 1.0–1.0) and of the repeat partial hepatectomy group 5.0 days (IQR 4.0–6.0) (*p* = 0.009). Margin size was <5 mm in 14.8% of tumors in the resection group and in 5.1% of tumors in the thermal ablation group.

### 3.3. Complications

No difference in complication rate was found between repeat thermal ablation and repeat partial hepatectomy (*p* = 0.063) (Table 3). Total complication rate was 21.8% (27/124 procedures), of which 19.2% (19/99 procedures) in the repeat thermal ablation group and 32.0% (8/25 procedures) in the repeat resection group. Two grade 4 complications were reported; one admission to the intensive care unit for respiratory insufficiency due to pneumonia (repeat resection group), and one patient suffered from intestinal wall injury resulting in colostomy (repeat thermal ablation group).

### 3.4. Local Tumor Progression-Free Survival

LTP was reported at follow-up in 25 out of 224 tumors (11.2%); 18/170 (10.6%) in the repeat thermal ablation group and 7/54 (13.0%) in the repeat resection group (Figure 2). Overall crude comparison between the two groups showed no significant difference in LTPFS (*p* = 0.959). Overall, 1-, 3- and 5-year LTPFS was 92.8%, 84.0% and 84.0%. The 1-, 3- and 5-year LTPFS was 91.6%, 85.8% and 85.8%, respectively, for the thermal ablation group and 96.1%, 81.4% and 81.4% for the repeat resection group. Univariable analysis identified three potential confounders: initial CRLM diagnosis (synchronous vs. metachronous; *p* = 0.002), time between initial treatment and diagnosis of recurrence (*p* = 0.003), and number of recurrent metastases (*p* = 0.016). These variables were included in multivariable analysis to analyze whether potential confounders associated with the two treatment groups influenced LTPFS (Appendix A). Only the variable time between initial treatment and diagnosis of recurrence proved a significant confounder in multivariable analysis (*p* = 0.001). After adjusting for this confounder corrected HR for LTPFS after repeat thermal ablation was 1.486 (95% CI, 0.594–3.714; *p* = 0.397).

### 3.5. Distant Progression-Free Survival

Ninety of 136 patients (66.2%) developed distant progression at follow-up with a median time to distant progression of 9.7 months (Figure 3). Following repeat thermal ablation and repeat resection, distant progression rate was 66.0% (66/100 patients) and 66.7% (24/36 patients), respectively. Overall, 1-year DPFS was 44.6%, 3-year DPFS was 24.7% and 5-year DPFS was 19.8%. The 1-, 3- and 5-year DPFS were, respectively, 44.4%, 24.0% and 19.8% for the thermal ablation group and 44.7%, 26.6% and 21.3% for the repeat resection group. No difference in DPFS was found in crude comparison (*p* = 0.803). Univariable analysis identified age (*p* = 0.092), initial CRLM diagnosis (synchronous vs. metachronous; *p* = 0.089), time between initial treatment and diagnosis recurrence (*p* = 0.032) and size of largest recurrent metastasis (*p* = 0.008) as potential confounders. Of these parameters, size of largest recurrent metastasis (*p* = 0.002) and time between initial treatment and diagnosis recurrence (*p* = 0.016) proved significant in multivariable analysis (Appendix A). After adjusting for these confounders, corrected HR was 1.024 (95% CI, 0.545–1.922; *p* = 0.942).

### 3.6. Overall Survival

Overall median OS as well as median OS of the repeat thermal ablation group was 54.4 months, whereas median OS of the repeat resection group was 49.2 months (Figure 4). During follow-up, a total of 46/136 patients (33.8%) died, 14/36 (38.9%) in the repeat resection group and 32/100 (32.0%) in the repeat thermal ablation group. The crude overall comparison of OS between the two groups revealed no significant difference (*p* = 0.927). Overall, 1-year OS was 97.5%, 3-year OS was 66.5% and 5-year OS was 44.1%. The 1-, 3- and 5-year OS were, respectively, 98.9%, 62.6% and 42.3% for the thermal ablation group and 93.8%, 74.5% and 49.3% for the repeat resection group. After identifying the association of comorbidities (*p* = 0.038) and primary tumor location (*p* = 0.083) with OS in univariable analyses, the variables were included in multivariable analysis to analyze their potential confounding influence on OS (Table 4). After adjusting for the confounder comorbidities (*p* = 0.038), corrected HR was 0.986 (95% CI, 0.517–1.881; *p* = 0.966). Subgroup analyses revealed no heterogeneous treatment effects according to patient, initial and repeat local treatment characteristics (Figure 5).

## 4. Discussion

Repeat partial hepatectomy of recurrent new CRLM was not statistically different from repeat thermal ablation with regard to crude overall comparison of OS (*p* = 0.927), complications (*p* = 0.063), LTPFS (*p* = 0.959) and DPFS (*p* = 0.803). Further quantification of OS, LTPFS and DPFS, after accounting for potential confounders, demonstrated concordant results for OS (HR, 0.986; 95% CI, 0.517–1.881; *p* = 0.966), LTPFS (HR, 1.486; 95% CI, 0.594–3.714; *p* = 0.397) and DPFS (HR, 1.024; 95% CI, 0.545–1.922; *p* = 0.942). Subgroup analyses identified no heterogeneous treatment effects according to patient, initial and repeat local treatment characteristics. 

Notably, length of hospital stay was longer in the repeat resection group compared to the repeat thermal ablation group (*p* = 0.009). Therefore, in addition to outcomes reported of thermal ablation versus partial hepatectomy for the initial local treatment of CRLM [10], this study no longer validates repeat partial hepatectomy as the only curative intent local treatment option for recurrent CRLM. The results even suggest that thermal ablation might be favored for small-size recurrent lesions suitable for both resection and ablation [7], given the lower invasiveness [30], lower costs [36] and reduced hospital stay when compared to surgery.

As a result of strict follow-up protocol after initial local treatment, new recurrent CRLM are detected relatively fast and therefore we observed merely small-sized recurrent metastases. In accordance with the presented results, the multidisciplinary COLLISION trial expert panel recommended thermal ablation as standard of care to treat small-size recurrent CRLM [7], because percutaneous thermal ablation is unaffected by post-surgical adhesions and a reduced liver volume [29,30].

Previous research on outcomes of repeat partial hepatectomy and repeat thermal ablation support our findings [37,38,39,40]. In the past, most studies analyzed survival outcomes of repeat resection compared to initial local treatment or to palliative chemotherapy. Yet, Dupré et al. analyzed well-matched patient groups with liver-limited recurrence after initial liver resection, treated with either repeat thermal ablation or resection [37]. No differences in median OS were found (both 33.3; 95% CI, 28–54.7 months) and the reduction in length of hospital stay (1 versus 5 days; *p* < 0.001) and lower rates of post-procedural complications (12.1% versus 38.7%; *p* = 0.021).

In contradiction to our results, Dupré et al. found inferior overall progression free survival for repeat partial hepatectomy compared to thermal ablation (10.2 versus 4.3 months; *p* = 0.002) [37]. One explanation can be the suboptimal comparison between pathology reports following partial hepatectomy and follow-up imaging exams following ablation. Dupré et al. did not take imaging based on recurrences following plane resections, for presumed R0 resections into account, nor did they compare A0 ablations, based on cross-sectional imaging directly after the procedure, with R0 resections, based on pathology reports. Nonetheless, even if in some centers the LTP rates following ablation are slightly higher than following partial hepatectomy, this does not automatically favor repeat surgery, given the relative ease to repeat thermal ablation and given the fact that it does not result in a worse oncological outcome [40].

Over the years, multiple improvements in ablative techniques, such as computed tomography hepatic arteriography (CTHA) guidance of percutaneous ablation, and developments in image fusion and navigation systems have resulted in increased tumor visualization with accurate needle tracking and positioning, and reduced complication rates [41,42]. By using image fusion and prediction of peri-ablational safety margins, technical success (A0 ablations) can be established and important prognosticators of LTP—safety margins of at least 5mm, and preferably >10mm—can be achieved [43,44,45,46,47]. All recent and future improvements are ultimately contributing to enhanced local tumor control and LTPFS. The prospect of rapidly improving techniques even further advocates repeat thermal ablation in patients with recurrent CRLM. However, the results of the recent OSLO-COMET randomized controlled trial (RCT) showing advantages of laparoscopic over open resection in complications (*p* = 0.021) and length of hospital stay (*p* < 0.001) should be taken into account [48].

Strengths of this study were the relatively high number of patients and tumors, which allowed sufficiently powered statistical analyses. Limitations are mainly inherent to the nonrandomized study design, considering that cohort studies are prone to selection bias and confounding. As analysis of OS, DPFS and LTPFS was conducted using Cox proportional hazards regression models, accounting for potential confounders in multivariable analysis, and subgroup analyses were performed to assess heterogeneous treatment effects according to patient, initial and repeat local treatment characteristics, risk of residual confounding is limited. An important limitation is that the MSI, RAS- and BRAF-mutation status were not routinely determined, therefore, these potential confounders could lead to residual bias. The long duration of the study may have caused underreporting of complications in the repeat partial hepatectomy group (11 patients missing), which may explain that no significant difference in complication rate was reported in this study compared to previous series [37]. Furthermore, choice of treatment and patient selection was based on local expertise, determined by multidisciplinary tumor board evaluations, preserving selection bias. In addition, the long study duration with gradual changes in indications for repeat local treatment, could have led to population bias. Nonetheless, no difference in patient characteristics between the two groups was identified. Furthermore, the thermal ablation techniques used in this study do not represent all contemporary, global thermal ablation techniques. 

## 5. Conclusions

To conclude, in this AmCORE based study repeat partial hepatectomy was not statistically different from repeat thermal ablation with regard to OS, DPFS, LTPFS and complications. Length of hospital stay favored repeat thermal ablation over repeat partial hepatectomy. Thermal ablation should be considered a valid and less invasive alternative to partial hepatectomy for small-size (0–3 cm) recurrent new CRLM, while the eagerly awaited results of the phase III prospective randomized controlled COLLISION trial (NCT03088150) should provide definitive answers regarding surgery versus thermal ablation for CRLM.

## Figures and Tables

**Figure 1 cancers-13-02769-f001:**
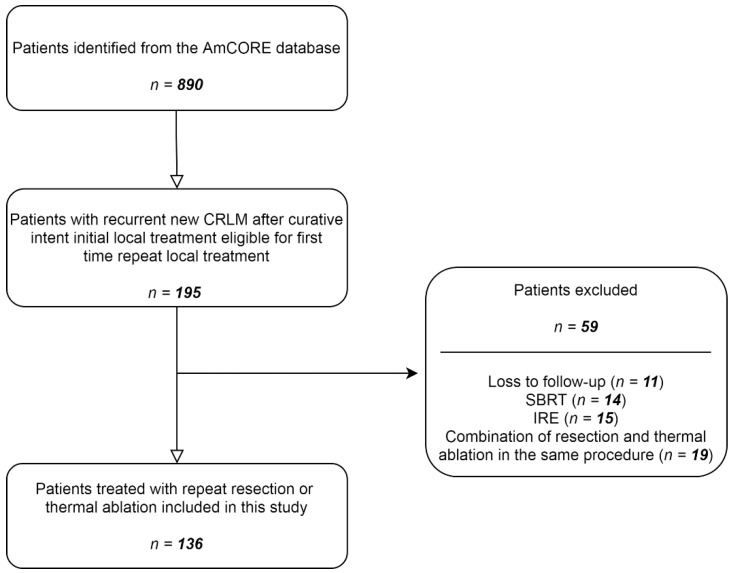
Flowchart of included and excluded patients.

**Figure 2 cancers-13-02769-f002:**
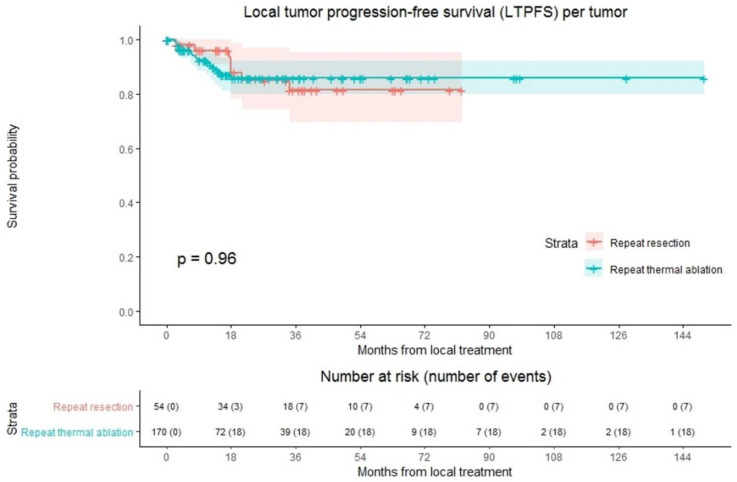
Kaplan–Meier curves of local tumor progression-free survival (LTPFS) per tumor after repeat resection (red) and repeat thermal ablation (green). Numbers at risk (number of events) are per tumor. Overall comparison log-rank (Mantel-Cox) test, *p* = 0.959. Death without local tumor progression (LTP; competing risk) is censored.

**Figure 3 cancers-13-02769-f003:**
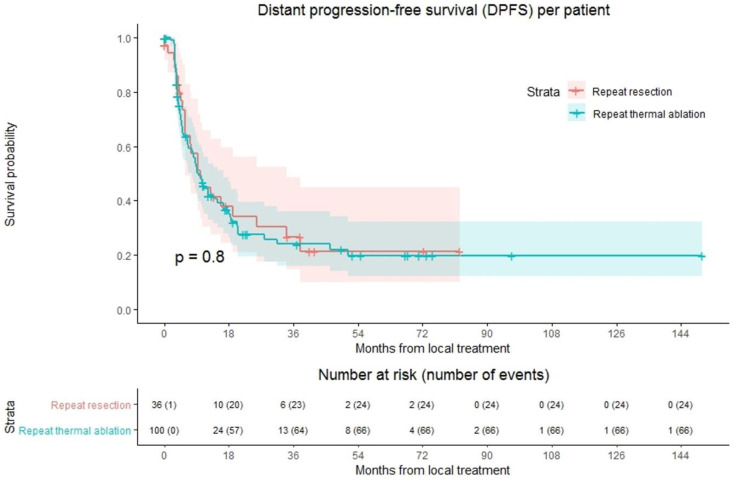
Kaplan–Meier curves of distant progression-free survival (DPFS) per patient after repeat resection (red) and repeat thermal ablation (green). Numbers at risk (number of events) are per patient. Overall comparison log-rank (Mantel-Cox) test, *p* = 0.803. Death without distant progression (competing risk) is censored.

**Figure 4 cancers-13-02769-f004:**
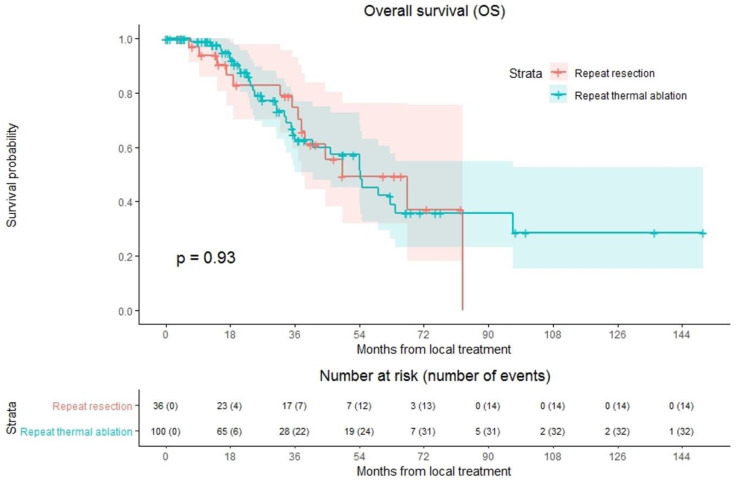
Kaplan–Meier curves of overall survival (OS) after repeat resection (red) and repeat thermal ablation (green). Numbers at risk (number of events) are per patient. Overall comparison log-rank (Mantel-Cox) test, *p* = 0.927.

**Figure 5 cancers-13-02769-f005:**
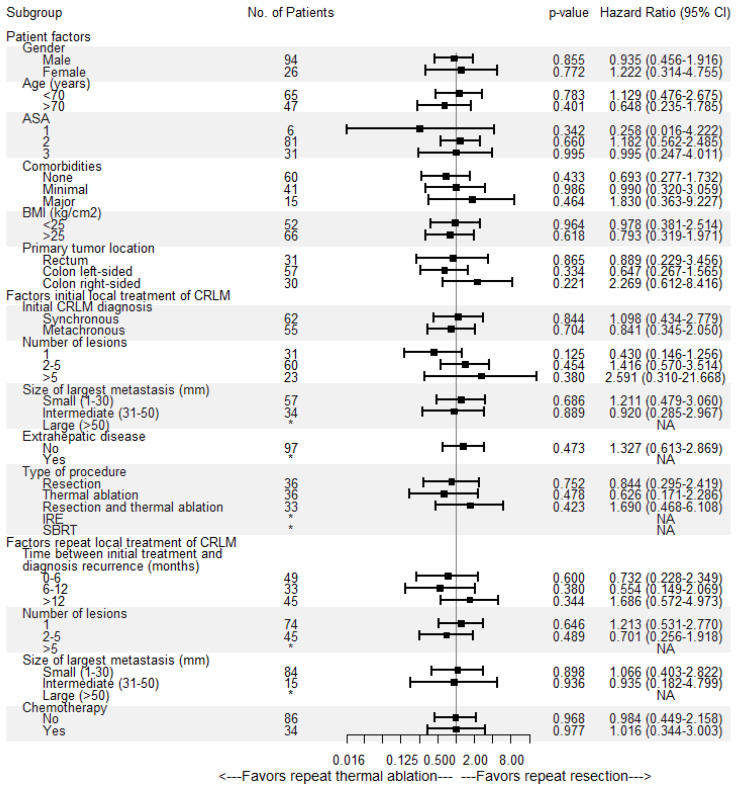
Univariable subgroup Cox regression analyses of repeat resection versus repeat thermal ablation associated with overall survival (OS). No = number, CI = confidence interval, ASA = American Society of Anesthesiologists score, BMI = body mass index, * = insufficient subgroup size for each treatment group, NA = not available.

**Table 1 cancers-13-02769-t001:** Baseline characteristics at recurrent CRLM.

		Total	Repeat Thermal Ablation Group	Repeat Resection Group	*p*-Value
Number of patients		136	100 (73.5)	36 (26.5)	
Number of tumors		224	170 (75.9)	54 (24.1)	
**Patient Characteristics**					
Gender	Male	104 (76.5)	79 (79.0)	25 (69.4)	
	Female	32 (23.5)	21 (21.0)	11 (30.6)	0.259 ^a^
Age (years) *		66.0 (10.9)	66.9 (11.4)	63.3 (9.1)	0.092 ^c^
ASA physical status	1	8 (5.9)	7 (7.0)	1 (2.8)	
	2	93 (68.4)	67 (67.0)	26 (72.2)	
	3	35 (25.7)	26 (26.0)	9 (25.0)	0.632 ^b^
Comorbidities	None	67 (49.3)	47 (47.0)	20 (55.6)	
	Minimal	49 (36.0)	38 (38.0)	11 (30.6)	
	Major	20 (14.7)	15 (15.0)	5 (13.9)	0.663 ^b^
BMI (kg/cm^2^) *		26.1 (4.2)	26.4 (4.2)	25.0 (4.0)	
	Missing	3	0	3	0.094 ^c^
Primary tumor location	Rectum	33 (24.3)	22 (22.0)	11 (30.6)	
	Colon left-sided	67 (49.3)	50 (50.0)	17 (47.2)	
	Colon right-sided	36 (26.5)	28 (28.0)	8 (22.2)	0.556 ^b^
**Characteristics Initial Local Treatment of CRLM**					
Initial CRLM diagnosis	Synchronous	69 (51.9)	51 (52.6)	18 (50.0)	
	Metachronous	64 (48.1)	46 (47.4)	18 (50.0)	0.847 ^a^
	Missing	3	3	0	
Number of tumors	1	38 (27.9)	26 (26.0)	12 (33.3)	
	2–5	65 (47.8)	45 (45.0)	20 (55.6)	
	>5	33 (24.3)	29 (29.0)	4 (11.1)	0.099 ^b^
Size of largest metastasis (mm)	Small (1–30)	75 (62.5)	57 (61.3)	18 (66.7)	
	Intermediate (31–50)	36 (30.0)	28 (30.1)	8 (29.6)	
	Large (>50)	9 (7.5)	8 (8.6)	1 (3.7)	0.681 ^b^
	Missing	16	7	9	
Extrahepatic disease	No	111 (92.5)	82 (93.2)	29 (90.6)	
	Yes	9 (7.5)	6 (6.8)	3 (9.4)	0.699 ^a^
	Missing	16	12	4	
Type of procedure	Resection	44 (32.4)	27 (27.0)	17 (47.2)	
	Thermal ablation	43 (31.6)	35 (35.0)	8 (22.2)	
	Resection and thermal ablation	46 (33.8)	37 (37.0)	9 (25.0)	
	IRE	2 (1.5)	1 (1.0)	1 (2.8)	
	SBRT	1 (0.7)	0 (0.0)	1 (2.8)	0.057 ^b^
**Characteristics Repeat Local Treatment of CRLM**					
Time between initial treatment and diagnosis recurrence (months) *		6.9 (4.0–13.4)	6.4 (4.0–10.4)	12.2 (3.7–21.3)	0.056 ^d^
Number of tumors	1	85 (62.5)	59 (59.0)	26 (72.2)	
	2–5	50 (36.8)	40 (40.0)	10 (27.8)	
	>5	1 (0.7)	1 (1.0)	0 (0)	0.337 ^b^
Size of largest metastasis (mm)	Small (1–30)	100 (84.7)	80 (85.1)	20 (83.3)	
	Intermediate (31–50)	16 (13.6)	13 (13.8)	3 (12.5)	
	Large (>50)	2 (1.7)	1 (1.1)	1 (4.2)	0.572 ^b^
	Missing	18	6	12	
Chemotherapy	No	98 (72.1)	71 (71.0)	27 (75.0)	
	Yes	38 (27.0)	29 (29.0)	9 (25.0)	0.829 ^a^

Values are reported as number of patients (%), * = continuous variables reported as mean (standard deviation; SD) or median (interquartile range; IQR), ^a^ = Fisher’s exact test, ^b^ = Pearson chi-square, ^c^ = independent t-test, ^d^ = Mann–Whitney U test, ASA = American Society of Anesthesiologists score, BMI = body mass index.

**Table 2 cancers-13-02769-t002:** Treatment characteristics of repeat local treatment.

		Repeat Thermal Ablation Group *n* = 100	Repeat Resection Group *n* = 36
Type of repeat thermal ablation	RFA	40 (40.0)	-
	Le Veen^TM^	35 (35.0)	
	Cool-tip^TM^	4 (4.0)	
	Others	1 (1.0)	
	MWA	60 (60.0)	-
	Emprint^TM^	46 (46.0)	
	Covidien Evident^TM^	5 (5.0)	
	Others	9 (9.0)	
Type of repeat resection	Minor (<3 segments)	-	34 (97.1)
	Major (≥3 segments)	-	1 (2.9)
	Missing		2
Approach	Open	17 (17.2%)	28 (84.8)
	Laparoscopic	0 (0.0)	5 (15.2)
	Percutaneous	82 (82.2%)	-
	Missing	1	3

Values are reported as number of patients (%), RFA = radiofrequency ablation, MWA = microwave ablation.

**Table 3 cancers-13-02769-t003:** Complications of repeat local treatment (CTCAE) [33].

Grade	Total	Repeat Thermal Ablation Group *n* = 100	Repeat Resection Group *n* = 36	*p*-Value
None	97 (78.2)	80 (80.8)	17 (68.0)	0.063 ^b^
Grade 1	8 (6.5)	8 (8.1)	NR	
Grade 2	8 (6.5)	4 (4.0)	4 (16.0)	
Grade 3	9 (7.3)	6 (6.1)	3 (12.0)	
Grade 4	2 (16)	1 (1.0)	1 (4.0)	
Grade 5	NR	NR	NR	
Missing	12	1	11	

Values are reported as number of patients (%), NR = not reported, ^b^ = Pearson chi-square.

**Table 4 cancers-13-02769-t004:** Univariable and multivariable cox regression analysis to detect potential confounders associated with overall survival (OS). After removal of primary tumor location and adjusting for the confounder comorbidities, corrected HR of repeat local treatment was 0.986 (95% CI, 0.517–1.881; *p* = 0.966).

		Univariable Analysis		Multivariable Analysis	
		HR (CI)	*p*-Value	HR (CI)	*p*-Value
Repeat local treatment	Repeat resection	Reference	0.927	Reference	0.966
	Repeat thermal ablation	0.971 (0.515–1.831)		0.986 (0.517–1.881)	
**Patient-Related Factors**					
Gender	Male	Reference	0.593		
	Female	0.826 (0.409–1.668)			
Age (years)		1.027 (0.994–1.062)	0.114		
ASA physical status	1	Reference	0.177		
	2	3.790 (0.894–16.078)			
	3	2.979 (0.644–13.780)			
Comorbidities	None	Reference	0.038	Reference	0.038
	Minimal	1.615 (0.853–3.061)		1.618 (0.850–3.079)	
	Major	2.940 (1.264–6.838)		2.936 (1.258–6.848)	
BMI (kg/cm^2^)		0.978 (0.906–1.056)	0.570		
Primary tumor location	Rectum	Reference	0.084	Reference	0.060
	Colon left-sided	0.902 (0.434–1.877)		0.879 (0.421–1.835)	
	Colon right-sided	1.918 (0.862–4.268)		2.002 (0.890–4.503)	
**Factors Regarding Initial Local Treatment of CRLM**					
Initial CRLM diagnosis	Synchronous	Reference	0.778		
	Metachronous	0.917 (0.503–1.672)			
Number of tumors	1	Reference	0.618		
	2–5	0.906 (0.465–1.764)			
	>5	0.663 (0.287–1.535)			
Size of largest metastasis (mm)	Small (1–30)	Reference	0.349		
	Intermediate (31–50)	0.864 (0.438–1.706)			
	Large (>50)	0.333 (0.075–1.478)			
Extrahepatic disease	No	Reference	0.250		
	Yes	0.311 (0.042–2.277)			
Type of procedure	Resection	Reference	0.798		
	Thermal ablation	1.360 (0.669–2.765)			
	Resection andthermal ablation	0.867 (0.413–1.822)			
	IRE	1.128 (0.147–8.645)			
	SBRT	*			
**Factors regarding repeat local treatment of CRLM**					
Time between initial treatment and diagnosis recurrence (months)		1.001 (0.980–1.022)	0.943		
Number of tumors	1	Reference	0.620		
	2–5	1.350 (0.740–2.464)			
	>5	*			
Size of largest metastasis (mm)	Small (1–30)	Reference	0.251		
	Intermediate (31–50)	1.795 (0.812–3.971)			
	Large (>50)	2.092 (0.481–9.103)			
Chemotherapy	No	Reference	0.825		
	Yes	1.071 (0.582–1.970)			

HR = hazard ratio, CI = 95% confidence interval, ASA = American Society of Anesthesiologists score, BMI = body mass index, * = insufficient subgroup size for each treatment group.

## Data Availability

The data presented in this study are available on request from the corresponding author.

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
