# Peer review of "Thermal Ablation Compared to Partial Hepatectomy for Recurrent Colorectal Liver Metastases: An Amsterdam Colorectal Liver Met Registry (AmCORE) Based Study"

_cancers, 2021, doi:10.3390/cancers13112769_

Round 1
Reviewer 1 Report
This article compares the outcomes of patients bearing reccurrent CRLM treated either by surgery or thermal ablation. This publication adresses a frequent and important question based on a large prospectively maintained unicenter datbase from an expert center.
The paper is very well written all along the manuscript. Statistics looks adequate however, some relevant informations are missing:
The biology of tumours in each group are nt described, RAS mutation rate in each group as well as MSI can subtantially influence survival and recucrence rate.
tecchniques are not described at all in each group including quality control criteria: R0 vs R1 for surgery, A0 A1 for ablation if early CT is done after ablation should be given as well. In the same field of information, surgery is not described at all: we do not know if anatomical resection is used or metastasectomy. Same level of information for ablation is missing.
The tumour size is not described in this publication. This might explain in group 1 (1-30mm) the good results of ablation vs surgery if this group has a low median size below 10mm for instance.
Discussion is fine and well balanced.
I strongly recommend to accept tis publication that will bring substantial infrmation in afield where belief is more proeminent than scientific evidence
Reviewer 2 Report
The paper is of interest but some major concerns are present:
- first of all, conclusion are drawn based on a retrospective analysis of different cohorts. This is not a randomized study, so you can't conclude it. Please clearly state this feature of your study and consider a possibile randomized prospective trial as a future development of your study
- tables are really problematic to be read, sometimes they are divided between 2 pages and not easy understandable. Please reduce data, make tables smaller and easier
- survival graphs and strata are also not clear. Please improve them
Round 2
Reviewer 2 Report
The paper has improved, now suitable for publication